# Simulation of Low-Temperature Oxidation and Combustion of N-Dodecane Droplets under Microgravity Conditions

**Sergey M. Frolov** [1,2,*] and **Valentin Y. Basevich** [1]

1   Department of Combustion and Explosion, Semenov Federal Research Center for Chemical Physics of the Russian Academy of Sciences, Moscow 119991, Russia
2   Institute of Laser and Plasma Technologies, National Research Nuclear University MEPhI (Moscow Engineering Physics Institute), Moscow 115409, Russia
*   Correspondence: smfrol@chph.ras.ru

**Abstract:** Fires are considered among the most dangerous accidents on manned spacecraft. That is why several programs of combustion experiments were implemented at the International Space Station (ISS) since 2008. In the experiments with n-heptane and n-dodecane droplet combustion, a new phenomenon was discovered, namely, the phenomenon of the radiative extinction of a burning droplet with subsequent multiple flashes of flame. In this paper, n-dodecane droplet ignition, combustion, radiative extinction, and subsequent low-temperature oxidation with multiple flashes of cool, blue, and hot flames under microgravity conditions are studied computationally. The mathematical model takes into account multiple elementary chemical reactions in the vicinity of a droplet in combination with heat and mass transfer in liquid and gas, heat release, convection, soot formation, and heat removal by radiation. The model is based on the non-stationary one-dimensional differential equations of the conservation of mass and energy in liquid and gas phases with variable thermophysical properties within the multicomponent diffusion concept in the gas phase. Calculations confirm the important role of the soot shell formed around the droplet and low-temperature reactions in the phenomenon of droplet radiative extinction with multiple flame flashes in the space experiment at the ISS. Calculations reveal the decisive role of the blue flame, arising due to the decomposition of hydrogen peroxide, in the multiple flame flashes. Calculations with forced ignition of the droplet reveal the effect of the ignition procedure on droplet evolution in terms of the timing and the number of cool, blue, and hot flame flashes, as well as in terms of the combustion rate constant of the droplet. Calculations with droplet self-ignition reveal the possible existence of new modes of low-temperature oxidation of droplets with the main reaction zone located very close to the droplet surface and with only partial conversion of fuel vapor in it.

**Keywords:** droplet combustion; n-dodecane; space experiment; microgravity; radiative extinction; low-temperature oxidation; mathematical modeling; detailed kinetic mechanism



## 1. Introduction

Fires are considered among the most dangerous accidents on manned spacecraft. Despite the microgravity, combustion products from the flames quickly spread in a confined space due to their thermal expansion. In the compartments of the spaceship, the composition of the atmosphere is disturbed, which carries the risk of poisoning the crew members, and the intense thermal effect on the onboard equipment can lead to an escalation of the process [1–3]. Of course, when designing spaceships and drawing up the "rules of stay" for the crew, everything possible is performed to exclude such accidents. However, the measures taken, as a rule, are based on the knowledge obtained in laboratories in the conditions of terrestrial gravity and on imaginary scenarios for the development of emergency situations. That is why the risk of carrying out a space experiment (SE) program for the ignition and combustion of various materials was considered justified by experts [4].

In 2008–2012, the US space agency NASA conducted the Flame Extinguishment (FLEX) SE on the International Space Station (ISS) [4]. In this SE, a new, previously unobserved phenomenon of radiative extinction of a spherical hot flame around large (2–5 mm in diameter) single droplets of individual hydrocarbon fuels (methanol, n-heptane) with their subsequent low-temperature "cool-flame" (according to the terminology of [4,5]) combustion under microgravity conditions [5–8]. Such low-temperature combustion of droplets without a visible flame, i.e., flameless combustion, was sometimes accompanied by multiple flashes of a high-temperature ("hot") flame.

In February–October 2017, another space experiment was carried out on the ISS as part of the cooperation program between NASA and the Russian Space Agency Roscosmos, which was called Cool Flame Investigation (CFI) by NASA and Zarevo by Roscosmos. The objective of the experiment was to study the combustion of single droplets of heavy hydrocarbon fuels of a complex structure (n-dodecane and its isomer, as well as farnesane (2,6,10 trimethyldodecane)) under microgravity conditions and to check whether they are inherent in the phenomena observed in the FLEX SE. American specialists designed and manufactured the experimental setup, which was delivered by a cargo ship to the American segment of the ISS. There, in microgravity conditions, Roscosmos cosmonaut Oleg Novitsky and NASA astronaut Peggy Whitson commissioned work on the installation and prepared it for operation under the remote control of ground services.

The installation included a sealed combustion chamber, auxiliary devices, and recording equipment (Figure 1) and operated in automatic mode. At the beginning of the experiment, the combustion chamber was filled with an oxidizer gas of a certain composition, for example, synthetic air. Then, with the help of two retractable needles, a droplet of liquid hydrocarbon, for example, n-dodecane, was formed in the center of the chamber, with a diameter of 2 to 5 mm. Then, the needles were removed, and an ignition source was supplied to the droplet: thin electrodes with a heated wire spiral. Immediately after ignition, the electrodes were removed, and the burning droplet floated in microgravity. The entire process, from the formation of the droplet to its complete disappearance, was recorded on several digital video cameras of different sensitivity and resolution, focused on the droplet. In addition, radiometers measured the radiation intensity of the droplet at different wavelengths. The operation of all recording equipment was synchronized, and the recorded information was transmitted via radio communication channels to the Space Flight Control Center in real time.

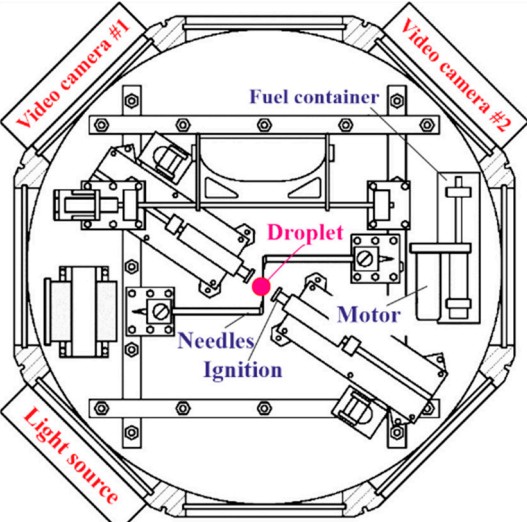

**Figure 1.** Combustion chamber, auxiliary devices, and recording equipment used in the Cool Flame Investigation—Zarevo space experiment at the ISS.

Hundreds of experiments were carried out in which the type of liquid hydrocarbon, the size of the droplets, the composition of the gas atmosphere, and the pressure of the oxidizer gas were varied. The experiments were carried out at room temperature of the oxidizer gas and droplets. After each experiment, the gas from the chamber was pumped out into a sealed container. The crew monitored the implementation of the experiment program, replaced the containers with gases and liquids, adjusted the equipment, and, if necessary, replaced the faulty units. All this required the greatest caution: any spills of flammable liquid or gas leaks and any glass fragments could immediately end up in the lungs of the crew members. Fortunately, there were no serious incidents.

The experiment showed the following. Immediately after ignition, a brightly glowing ball of flame formed around the droplet (frame 1 in Figure 2). After a few seconds, the intensity of the glow of the flame sharply decreased (frame 2), and the glow completely disappeared (frame 3). However, after some time (after a few seconds), a brightly glowing ball of flame again appeared around the droplet (frame 4), and then the flame disappeared again. Such flashes and extinctions of the flame around the droplet could occur several times during its life. Even if there were no flashes, the droplet still evaporated unexpectedly quickly. When conducting a similar experiment under the conditions of terrestrial gravity, nothing like this happened: a droplet suspended on a thin thread completely burned out in a bright luminous flame, only the flame of the droplet did not have the shape of a luminous ball, but an elongated shape, like a wax candle.

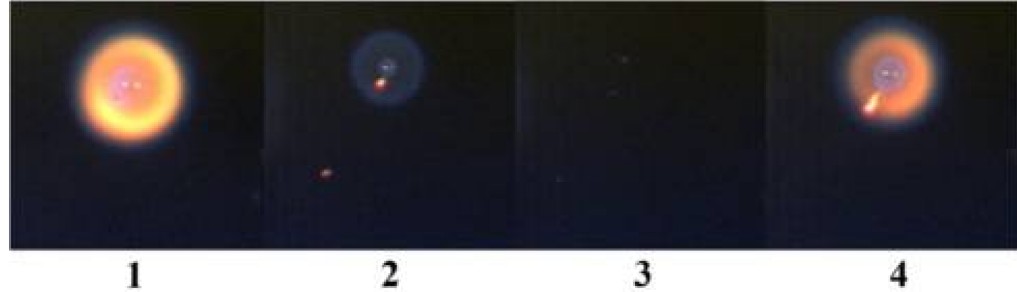

**Figure 2.** Extinction of the droplet (frames 1 and 2), flameless combustion of the droplet (frame 3), and repeated flash of the flame of the droplet (frame 4) under microgravity conditions.

A careful analysis of the video frames obtained with another, more sensitive camera showed that when a droplet burned in microgravity conditions, an interesting effect arose: between the flame and the droplet, a spherical layer of very fine soot particles thickened over time (Figure 3), the so-called "soot shell." The fact that soot is formed during the combustion of a droplet has long been known to science [9,10]. It is also known that soot is formed inside the flame, where the concentration of fuel vapors and temperature are high. Under terrestrial conditions, under the action of the Archimedes force, small particles of soot, formed near the droplet, are carried upwards with hot gases. In the SE, there is no Archimedes force, and soot particles accumulate around the drop. In our opinion, it is this effect that is the primary cause of the observed differences in the combustion of droplets on the ground and in microgravity.

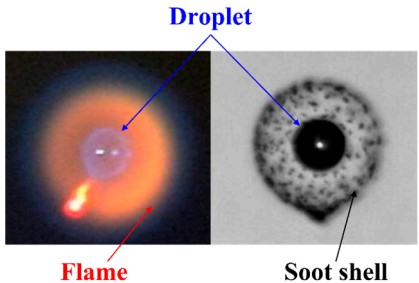

**Figure 3.** Formation of a soot shell around a burning droplet.

The soot shell, which thickens with time, surrounding the burning droplet in the SE, becomes a kind of thermal screen between the flame and the droplet. Soot particles take part in the heat flux from the flame to the droplet and radiate heat into the surrounding space. This leads to a decrease in the droplet evaporation rate and a decrease in the inflow of fuel vapor to the flame. As a result, the burning intensity of the droplet decreases, and the flame quenches. Under terrestrial conditions, the soot shell around the burning droplet is continuously carried away with hot gases and has a relatively low density, i.e., the ability of the soot shell to screen the heat flux from flame to droplet is less pronounced.

Why do repeated flashes of flame occur in the SE, and why does the droplet disappear much faster than expected even in the absence of the repeated flashes? To obtain the answers to these questions, there is a need for a deep knowledge of fuel oxidation chemistry and detailed simulation of droplet ignition and combustion.

It is known that n-dodecane, as a representative of the homologous series of normal alkanes, exhibits a multistage low-temperature self-ignition due to the existence of competing mechanisms of chain branching. The multistage ignition of homogeneous n-dodecane–air mixtures occurs as a sequence of cool, blue, and hot flames [11]. The acceleration of the reaction in the cool flame is a consequence of chain branching during the decomposition of alkyl hydroperoxide (here, $C_{12}H_{25}O_2H$) with the formation of a hydroxyl and an alkyloxy radical. The cool flame is detected as a very dim glow of excited formaldehyde $CH_2O$. A blue flame arises because of chain branching during the decomposition of hydrogen peroxide $H_2O_2$. The blue flame is detected as a bluish glow of both excited formyl CHO and excited formaldehyde. Importantly, the glow intensity of the blue flame is considerably higher than that of the cool flame. Hot flame is a consequence of a branched-chain reaction of atomic hydrogen with molecular oxygen. One of the most striking features of multistage self-ignition of normal alkanes is the existence of a region with a negative temperature coefficient (NTC) of the reaction rate; when at a higher initial temperature, the total ignition delay time is longer than that at a low temperature.

Methods of modern mathematical theories of combustion make it possible to calculate the characteristics of the self-ignition and laminar combustion of gaseous mixtures of n-alkanes on the basis of first principles, i.e., without introducing adjustment coefficients and to obtain a satisfactory agreement with experimental data. Such calculations are performed using detailed kinetic mechanisms (DKMs) of fuel oxidation and databases of thermophysical properties of substances (see, e.g., [12–17]).

As for the ignition and subsequent burning of droplets of normal alkanes, simulations of these processes started long ago with the use of overall kinetic mechanisms [18–20]. In [21], the self-ignition of single n-heptane droplets was simulated using the semi-empirical kinetic mechanism, which included both high- and low-temperature reactions simulating the NTC of the reaction rate in a homogeneous mixture. However, the authors of [21] revealed no NTC region while simulating the low-temperature self-ignition of droplets. Numerical simulations of the low-temperature self-ignition of single n-heptane droplets with the use of a semi-empirical mechanism containing 62 reactions were performed in [22]. In contrast to [21], in the calculations of [22], an NTC region was detected, but when compared to the experimental data, it was significantly shifted toward lower temperatures. In [23], the high-temperature self-ignition of single n-heptane droplets was simulated using a semi-empirical mechanism composed of 282 and 51 components. The main drawback of semi-empirical mechanisms is that their applicability to the specific conditions of calculations should be tested using experimental data. However, since experimental data are usually scarce, these mechanisms are often extrapolated to a region of governing parameters in which such tests have not been conducted.

The most reliable way of modeling self-ignition and combustion of droplets of heavy normal alkanes is to use a DKM. One of the first attempts at using detailed chemistry in droplet calculation was undertaken in [24]. In [25], on the basis of a DKM including 904 reactions and 168 components, simulations of the low-temperature self-ignition delays in a three-dimensional two-phase turbulent flow with monodisperse droplets of n-heptane were

performed. Owing to the large computational cost, the modeling of accompanying physical processes was significantly simplified: the temperature distribution over the droplet and the computational mesh, whose size was many times the diameter of the droplet, was assumed uniform. In fact, the authors of [25] simulated the gas-phase low-temperature self-ignition of n-heptane with some averaged parameters of the evaporation of the liquid. In [26], a numerical simulation of forced ignition and combustion of single n-heptane droplets in a 25%$O_2$ + 75%He atmosphere was performed with the use of the block of reactions from the DKM of n-decane oxidation containing 5000 reactions and involving 200 components. The focus was on the sensitivity of the conditions of extinction of the flame around the droplet to changes in the governing parameters of the problem. Note that, in none of the cited works and later works [27,28] on the mathematical modeling of the ignition and combustion of droplets of n-heptane, a blue flame was observed, and even the possibility of its existence was not discussed.

In [29], based on the mathematical model of droplet combustion [30] and the DKM of n-heptane oxidation and combustion [31], all the main features of the phenomenon observed in the FLEX SE were quantitatively reproduced, and the existence of new modes of low-temperature droplet combustion without the stage of hot flame was predicted. The only change in the model [30] and in the DKM was to take into account the emission of soot formed during combustion. It was shown in [29] that after hot flame extinction, the n-heptane drop could continue to evaporate because of the oxidation of fuel vapors with the repeated flashes of cool, blue, and hot flames rather than exceptionally the cool flame, as claimed in [4,5]. In [32], the same mathematical model was applied to large n-dodecane droplets using the DKM of n-dodecane oxidation and combustion [33] and taking into account soot emission, as was conducted in [34,35]. The calculations in [32] showed that, despite the differences in the physical nature of the ignition and combustion of homogeneous mixtures and droplets, the local chemical processes manifested themselves in the same way. Despite the kinetics of flame glow being absent in the DKM, it was assumed that the concentration of excited formyl and formaldehyde molecules was proportional to the concentration of these components in the gas phase. The analysis showed that the low-temperature reactions led to the appearance of both cool and blue flames.

In the present work, based on the model of droplet combustion and the DKM of n-dodecane oxidation combined with the microkinetic mechanism of soot formation, the specific features of forced ignition and self-ignition and combustion of large and small n-dodecane droplets under microgravity conditions were studied computationally. Particular attention is paid to the role of the blue flame in multiple flame flashes after radiative extinction of droplet combustion.

## 2. Materials and Methods

### 2.1. Mathematical Model

Consider a single-component fuel droplet in an oxidizing atmosphere using the following simplifying assumptions:

(1)    there is no natural and forced convection;
(2)    nonequilibrium effects are negligible;
(3)    the effects of thermal diffusion are insignificant;
(4)    the solubility of gas in a liquid is negligible.

In the absence of natural and forced convection, the droplet has a spherical shape. This allows one to consider the processes of droplet heating, evaporation, forced ignition/self-ignition, and combustion as spherically symmetric. It is assumed that the droplet occupies the region $0 < r < r_m$, where $r$ is the radial coordinate originating in the droplet center, and $r_m$ is the coordinate of the droplet surface (droplet radius). Hereafter, liquid and gas parameters are denoted by indices $d$ and $g$, respectively, whereas indices $m$ and 0 refer to the droplet surface and initial values of variables, respectively. The initial droplet radius is $r_{m0}$. The droplet radius changes with time $t$, i.e., the coordinate $r_m$ moves with speed $u_m = u_m(t)$. The radius $r_m$ can change due to two factors, namely, due to the liquid–vapor

phase transition on the droplet surface and due to the thermal expansion (compression) of the liquid caused by the dependence of the liquid density on temperature. Thus, the droplet radius $r_m(t)$ is determined from the law of conservation of liquid mass:

$$m_d(t) = 4\pi \int_0^{r_m(t)} \rho_d(r) r^2 dr$$

written in the form

$$\frac{dm_d}{dt} = 4\pi r_m^2 \rho_{d,m} u_{d,m}, \ m_d(0) = m_{d0}$$

where $\rho_{d,m}$ is the liquid density at the droplet surface, $u_{d,m}$ is the rate of change in the droplet radius due to the phase transition, which in the general case differs from $u_m$.

The mathematical model of heating, evaporation, self-ignition, and combustion of the droplet is then based on one-dimensional non-stationary differential equations of conservation of mass and energy in the liquid and gas phases with variable thermophysical properties. In formulating the problem, the concept of multicomponent diffusion in the gas phase is used. The model is constructed for the conditions of microgravity and constant pressure in the gas–droplet system [36].

The continuity equation for the liquid $(0 < r < r_m)$:

$$\frac{\partial \rho_d}{\partial t} + \frac{1}{r^2} \frac{\partial}{\partial r} \left( r^2 \rho_d u_d \right) = 0 \tag{1}$$

where $u_d$ is the velocity of the liquid.

The energy conservation equation for the liquid $(0 < r < r_m)$:

$$c_d \rho_d \frac{\partial T_d}{\partial t} + c_d \rho_d u_d \frac{\partial T_d}{\partial r} = \frac{1}{r^2} \frac{\partial}{\partial r} \left( \lambda_d r^2 \frac{\partial T_d}{\partial r} \right)$$

$$T_d(0, r) = T_{d0}, \ \left. \frac{\partial T_d}{\partial r} \right|_{r=0} = 0, \tag{2}$$

$$T_d(t, r_m) = T_g(t, r_m)$$

where $T_d(t, r)$ is the temperature of the liquid, $c_d(T_d)$ is the specific heat of the liquid, and $\lambda_d$ is the thermal conductivity of the liquid.

The equation for the mass concentration of liquid vapor at the droplet surface $(r = r_m)$:

$$Y_v = \frac{P_v}{P} \frac{W_v}{\overline{W}} \tag{3}$$

where $P$ is the pressure, $W$ is the molecular mass (the overbar denotes the average value), and index $v$ refers to the liquid vapor.

The continuity equation for the gas phase $(r_m < r < R)$:

$$\frac{\partial \rho_g}{\partial t} + \frac{1}{r^2} \frac{\partial}{\partial r} \left( r^2 \rho_g u_g \right) = 0$$

$$\rho_d \left( u_d - \frac{\partial r_m}{\partial t} \right) \bigg|_{r=r_m} = \rho_g \left( u_g - \frac{\partial r_m}{\partial t} \right) \bigg|_{r=r_m} \tag{4}$$

where $R$ is the external radius of the computational domain around the droplet, and the derivative $\partial r_m / \partial t = u_m$ determines the instantaneous velocity of the droplet surface due to both evaporation and thermal expansion.

The equation of continuity of the gaseous components $(r_m < r < R)$:

$$\rho_g \frac{\partial Y_j}{\partial t} + \rho_g u_g \frac{\partial Y_j}{\partial r} = \frac{1}{r^2} \frac{\partial}{\partial r} \left( \rho_g r^2 Y_j V_j \right) + \omega_{gj}$$

$$Y_j(0, r) = Y_{j0}, \ j = 1, 2, \ldots, N,$$

$$-\rho_d u_d \beta_i \big|_{r=r_m} = \rho_g Y_j \left( u_g - \frac{\partial r_m}{\partial t} \right) + \rho_g Y_j V_j \big|_{r=r_m},$$

$$\frac{\partial \overline{W} Y_j}{\partial r} \bigg|_{r=R} = 0, \ j = 1, 2, \ldots, N$$

(5)

where $V_j$ is the diffusion velocity of the $j$th component. The initial mass concentrations $Y_{j0}$ are specified in the form of uniform distributions in the gas phase. The rates of chemical reactions $\omega_{gj}$ and the coefficients $\beta_i$ are determined as

$$\omega_{gj} = W_{gj} \sum_{k=1}^{L} \left( v''_{j,k} - v'_{j,k} \right) A_k T_g^{n_k} \exp \left( -\frac{E_k}{RT_g} \right) \prod_{l=1}^{N} \left( \frac{Y_{gl} \rho_g}{W_{gl}} \right)^{v'_{l,k}},$$

$$\beta_i = 1 \text{ at } j = v$$

$$\beta_i = 0 \text{ at } j \neq v$$

where $A_k$, $n_k$, and $E_k$ are the pre-exponential factor, the temperature exponent, and activation energy for the $k$th reaction; $v'_{j,k}$ and $v''_{j,k}$ are the stoichiometric coefficients for the $j$th component in the case when it is a reactant and product in the $k$th reaction, respectively.

The equation for the diffusion velocity in the gas phase $(r_m < r < R)$:

$$\frac{\partial X_j}{\partial r} = \sum_{k=1}^{N} \left( \frac{X_j X_k}{D_{jk}} \right) (V_k - V_j)$$

(6)

where $X_j = Y_j \overline{W} / W_j$ is the mole fraction of the $j$th component in the mixture.

The equations of conservation of energy in the gas phase $(r_m < r < R)$:

$$c_{pg} \rho_g \frac{\partial T_g}{\partial t} + c_{pg} \rho_g u_g \frac{\partial T_g}{\partial r} = \frac{1}{r^2} \frac{\partial}{\partial r} \left( \lambda_g r^2 \frac{\partial T_g}{\partial r} \right) + \Omega - \sigma S_{\text{rad}} Y_s \rho_g T_g^4,$$

$$T_g(0, r) = T_{g0}(r), \ T_g(t, r_m) = T_d(t, r_m), \ \frac{\partial T_g}{\partial r} \bigg|_{r=R} = 0,$$

(7)

where $c_{pg} = c_{pg}(T_g)$, $\rho_g = \rho_g(p, T_g)$, and $\lambda_g = \lambda_g(p, T_g)$ are, respectively, the specific heat, density, and thermal conductivity of the gas mixture. The chemical source $\Omega$ in Equation (7) is given by

$$\Omega = \sum_{k=1}^{L} H_k A_k T_g^{n_k} \exp \left( -\frac{E_k}{RT_g} \right) \prod_{j=1}^{N} \left( \frac{Y_{gj} \rho_g}{W_{gj}} \right)^{v'_{l,k}}$$

where $H_k$ is the thermal effect of the $k$th chemical reaction. The last term in Equation (7) represents the heat loss due to soot radiation. Here, $\sigma$ is the Stefan–Boltzmann constant, $Y_s$ is the mass fraction of soot (soot is designated as C atom and is modeled by an equivalent gas with a molecular mass of carbon, 12 kg/kmol), and $S_{\text{rad}} = 6/(d_s \rho_s)$ is the specific surface area of conditional soot particles (here, $d_s$ is the conditional soot particle size, and $\rho_s$ is the soot density). If one assumes $d_s \sim 1$ nm and $\rho_s \approx 2000$ kg/m$^3$, then $S_{\text{rad}} \approx 3 \times 10^6$ m$^2$/kg.

The boundary condition of matching Equations (2) and (7) at the droplet surface $(r = r_m)$ for determining the droplet surface temperature $T_{d,m}$:

$$\lambda_d \frac{\partial T_d}{\partial r} - \frac{\rho_{d,m} u_{d,m} L_v}{W_v} = \lambda_g \frac{\partial T_g}{\partial r}$$

(8)

where $L_v$ is the latent heat of vaporization of the liquid.

The equation of state of an ideal gas for the gas phase:

$$\rho_g = \frac{P\overline{W}}{RT_g} \tag{9}$$

The condition of constant pressure:

$$P = const \tag{10}$$

The model thus formulated allows one to determine the spatial structure of the flow around the droplet and its evolution in time and calculate the time dependence of such characteristics of the problem as the droplet diameter $D = 2r_m$, droplet surface temperature $T_{d,m}$, the maximum gas temperature $T_m = T_{g,max}$ (flame temperature), as well as the integral mass fractions of all species in the gas phase, including soot, as:

$$I_{m,i}(t) = \frac{1}{m_{d0}} \int_{r_m}^{R} 2\pi \rho_g Y_i(t) r^2 dr, \ i = 1, 2, \ldots, N. \tag{11}$$

*2.2. Solution Procedure*

The system of Equations (1)–(10), supplemented by a database of the thermophysical properties of the components [36], was integrated numerically using a nonconservative implicit finite difference scheme and an adaptive moving grid. To calculate the chemical sources $\omega_{gj}$ and $\Omega$, the DKM [33] combined with the macrokinetic mechanism of soot formation [34,35] was used. The combined mechanism contained 84 species (including C) and 623 reversible and 4 irreversible (Table 1) reactions. The Arrhenius parameters in the mechanism of Table 1 were determined using the thoroughly tested DKM of soot formation [37]. In the DKM of [37], soot particle nuclei are formed in processes involving a stable polyaromatic molecule and a radical or two polyaromatic radicals. In the mechanism of Table 1, the role of soot precursor is attributed to acetylene $C_2H_2$.

**Table 1.** Macrokinetic mechanism of soot formation.

| Reaction | $A_k$, (L, mol, s) | $E_k/R$, K | $n_k$ |
|---|---|---|---|
| $C_2H_2 + C_2H_2 = C + C + C_2H_4$ | $2 \times 10^{16}$ | 40,000 | 0 |
| $C + CO_2 = CO + CO$ | $1 \times 10^{15}$ | 40,000 | 0 |
| $C + H_2O = H_2 + CO$ | $1 \times 10^{15}$ | 40,000 | 0 |
| $C + OH = HCO$ | $1 \times 10^{12}$ | 0 | 0 |

The calculation procedure consisted of successive approximations at each time step. An important feature of the algorithm was a complete linearization of the conditions of matching the solution at the droplet surface. The accuracy of the solution was continuously checked for compliance with the elemental balance of C and H atoms, as well as with energy balance. The maximum allowable deviation in the balances was 0.1%.

**3. Results and Discussion**

*3.1. Self-Ignition of Homogeneous Mixtures*

The DKM of [33] was validated earlier against the experimental data on self-ignition, laminar flame propagation, and counterflow diffusion flames for different alkane hydrocarbons up to n-hexadecane. As an example, Figures 4 and 5 compare the predicted and measured dependences of the ignition delay time, $t_i$, on temperature (Figure 4) and the laminar flame velocity, $u_n$, on the fuel-to-air equivalence ratio, $\Phi$ (Figure 5), for n-dodecane–air mixtures at different initial conditions in terms of the fuel-to-air equivalence ratio, temperature, and pressure. The mechanism describes both multistage low-temperature oxidation with cool and blue flames and hot flame. Figure 6 shows the calculated time histories of temperature during self-ignition of the homogeneous stoichiometric n-dodecane–air

mixture at two initial gas temperatures, $T_{g0} = 650$ K and 1000 K, and $P = 1$ bar with no droplet. The calculations are performed using CHEMKIN code.

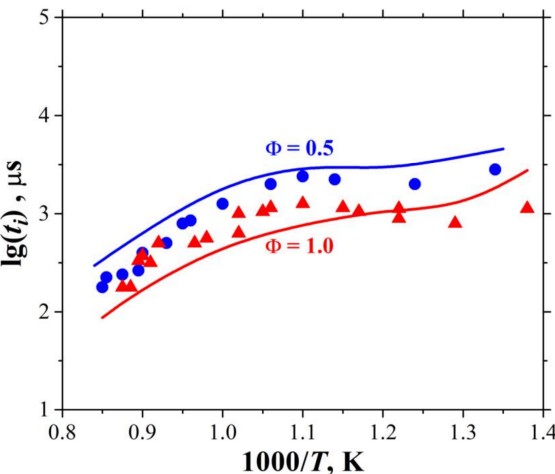

**Figure 4.** Calculated (curves) and measured (symbols) dependences of the ignition delay time on temperature for n-dodecane–air mixtures with $\Phi = 0.5$ and 1.0 at $P = 20$ bar.

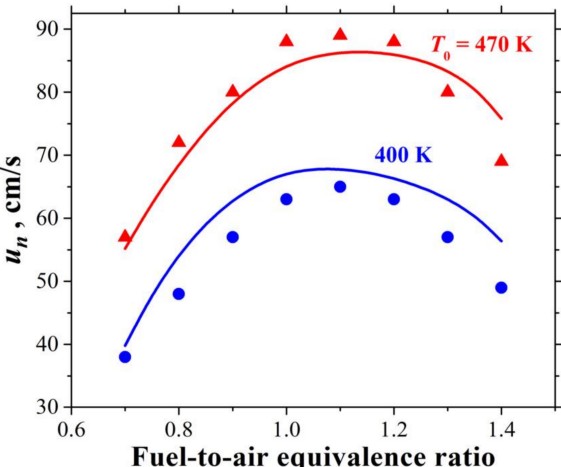

**Figure 5.** Calculated (curves) and measured (symbols) dependences of the laminar flame velocity on the fuel-to-air equivalence ratio for n-dodecane–air mixture at $P = 1$ bar and $T_0 = 400$ and 470 K.

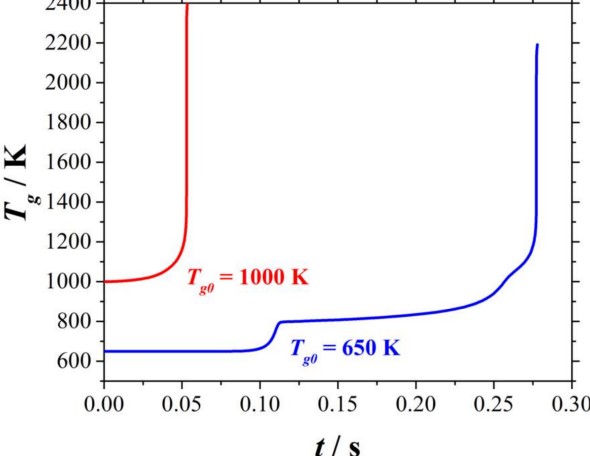

**Figure 6.** Time histories of gas temperature during self-ignition of the homogeneous stoichiometric n-dodecane–air mixture at $P = 1$ bar and $T_{g0} = 650$ K and 1000 K (no droplet).

The temperature evolution at $T_{g0} = 1000$ K is seen to be smooth (single-stage). At $T_{g0} = 650$ K, the typical low-temperature multistage evolution of temperature is observed. At a time of approximately 0.11 s, a temperature jump is seen, which corresponds to the appearance of a cool flame, whereas, at a time of approximately 0.25 s, a smooth wave is seen, which corresponds to the blue flame. It is known that the occurrence of the cool flame is due to the decomposition of alkyl peroxide $C_{12}H_{25}O_2H$ with the appearance of an additional hydroxyl OH, which accelerates the oxidation reaction. The occurrence of the blue flame is due to the decomposition of hydrogen peroxide $H_2O_2$ with the appearance of an additional hydroxyl OH accelerating the oxidation reaction. The occurrence of the cool and blue flames at $T_{g0} = 650$ K is clearly demonstrated in Figure 7, which shows the predicted time histories of $C_{12}H_{25}O_2H$, $H_2O_2$, and OH. The mass fraction of hydroxyl represents the overall rate of oxidation reaction. There are three peaks in the evolution of OH mass fraction: the first arises at a time of the maximum rate of $C_{12}H_{25}O_2H$ decomposition, the second arises at a time of the maximum rate of $H_2O_2$ decomposition, whereas the third (shown only partly) corresponds to the hot flame due to the chain branching reaction $H + O_2 = O + OH$.

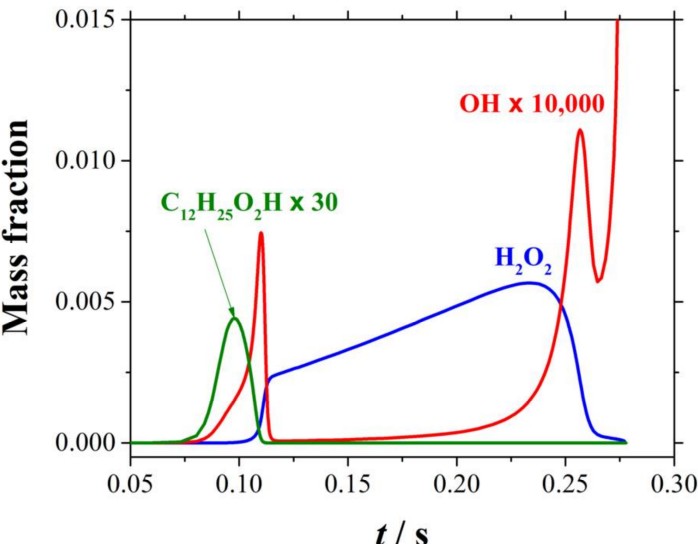

**Figure 7.** Time histories of peroxides $C_{12}H_{25}O_2H$ and $H_2O_2$ as well as hydroxyl OH during self-ignition of the homogeneous stoichiometric n-dodecane–air mixture at the initial temperature $T_{g0} = 650$ K and pressure $P = 1$ bar (no droplet).

*3.2. Droplet Combustion*

Figure 8 compares the calculated and measured time histories of the $D^2/D_0^2$ ratio for the combustion of an n-dodecane droplet with $D_0 \approx 4.3$ mm in the synthetic air with $X_{O2} = 0.21$ and $X_{N2} = 0.79$ at NPT conditions. In the ISS experiment, the droplet exhibited flame extinction at $t/D_0^2 \approx 0.55$ s/mm², whereas flame extinction in the calculation is registered at approximately $t/D_0^2 \approx 1$ s/mm². Despite this difference, the calculation predicts satisfactorily the droplet lifetime. Surprisingly, flame extinction does not lead to a significant change in the rate of droplet depletion in both experiment and calculation. To better understand the underlying physicochemical phenomena, we consider below the details of droplet forced ignition and combustion.

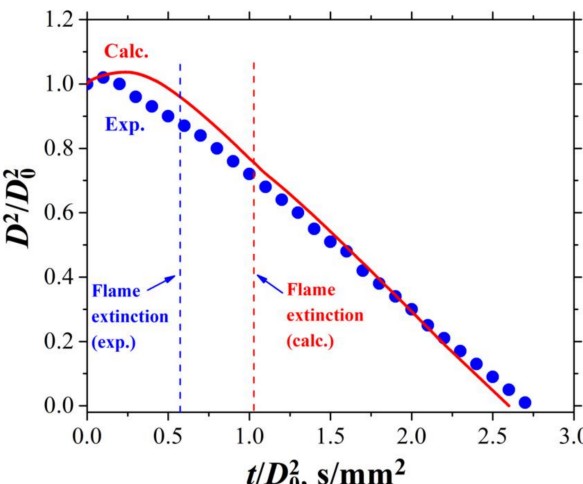

**Figure 8.** Calculated and measured time histories of the $D^2/D_0^2$ ratio for the combustion of an n-dodecane droplet with $D_0 \approx 4.3$ mm in the synthetic air with $X_{O2} = 0.21$ and $X_{N2} = 0.79$ at NPT conditions.

### 3.3. Forced Ignition of the Droplet

Consider the calculation results with forced ignition and combustion of the n-dodecane droplet 2.8 mm in initial diameter placed in the unconfined atmosphere of air at normal pressure and temperature (NPT) conditions. The procedure of the forced ignition of droplets used in experiments [4–8] is modeled in the calculations by placing a spherical layer with a high temperature around the droplet, which leads to the appearance of a hot flame. As an example, Figure 9 shows two variants of the initial temperature field $T_0(r)$. The initial droplet radius $r_{m0}$ and the width of the isothermal section, $L = 0.55$ and $0.53$ cm, with high temperature (2000 K), are marked here for the sake of convenience. The temperature is seen to smoothly decrease to the initial temperature of the ambient air. As it turned out, the heat content of the igniting layer, which varies due to the width of the isothermal section $L$, significantly affects further evolution of the droplet. Note that a similar procedure was used in calculations [29] for igniting a droplet of n-heptane. It appeared that for igniting a droplet of n-dodecane, slightly higher energy was required compared to a droplet of n-heptane.

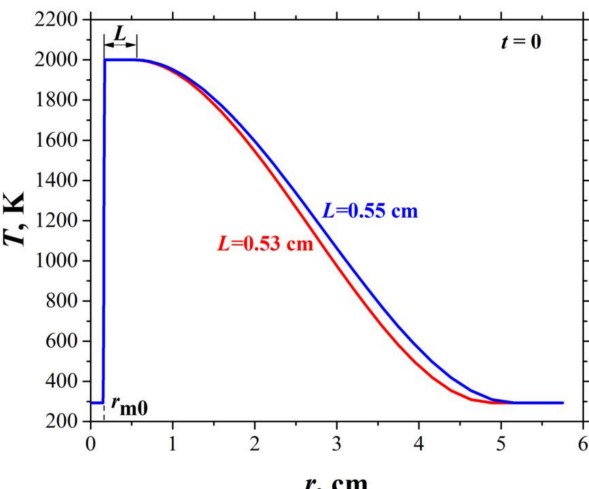

**Figure 9.** An example of setting the initial temperature fields $T_{l0}(r)$ and $T_{g0}(r)$ for the n-dodecane droplet with $D_0 = 2.8$ mm ($r_{m0} = 1.4$ mm) at NPT conditions.

Figure 10 shows the calculated time histories of the maximum gas temperature, $T_m$, and the normalized squared droplet diameter, $D^2/D_0^2$, for the forced ignition and combustion of the n-dodecane droplet with $D_0 = 2.8$ mm and $L = 0.55$ cm in air at NPT conditions. The initial increase in the $D^2/D_0^2$ ratio is caused by the domination of liquid heating and thermal expansion over liquid evaporation. The duration of this transient initial period is approximately 2 s. Thereafter, the $D^2/D_0^2$ ratio continuously decreases. The slope of the dashed line corresponds to the burning rate constant $k$ in the well-known linear $D^2$-law: $D^2 = D_0^2 - kt$. At $t > 0$, the maximum gas temperature $T_m$ around the droplet first gradually decreases from 2000 K to about 900 K. After a delay approximately equal to 1.22 s, the temperature begins to rise, leading to droplet ignition, which is accompanied by the temperature rise to 1890 K. Then, due to accumulation of soot accompanied by radiation losses, the high-temperature hot flame is extinguished, and the maximum gas temperature drops to $T_m = T_l \approx 900–1150$ K. This temperature range is shown in Figure 10 as a blue band enveloping low-temperature flashes of cool and blue flame.

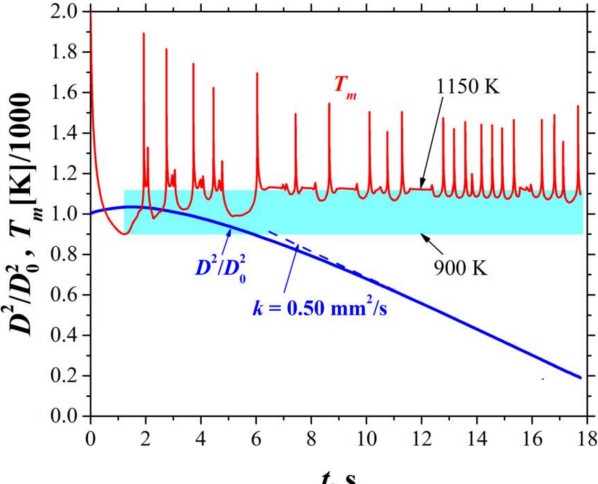

**Figure 10.** Calculated time histories of the maximum gas temperature, $T_m$, and the normalized squared droplet diameter, $D^2/D_0^2$, at the forced ignition and combustion of the n-dodecane droplet with $D_0 = 2.8$ mm and $L = 0.55$ cm in air at NPT conditions. The blue band envelopes the low-temperature flashes of cool and blue flames.

Figure 11 shows a fragment of calculated time histories of the maximum gas temperature, $T_m$, and the integral mass fractions of active species OH, O, and H immediately after the forced ignition of the n-dodecane droplet with $D_0 = 2.8$ mm and $L = 0.55$ cm in air at NPT conditions. The initial increase in the integral mass fractions of active species OH, O, and H is clearly seen. The emerging flame is maintained for approximately 0.5 ms, which is the time period considered in Figure 11, and extinguishes. The gas temperature $T_m = T_l \approx 1000–1150$ K attained after hot flame extinguishment is characteristic of the blue-flame low-temperature oxidation of n-dodecane caused by the decomposition of hydrogen peroxide (see Figure 4). Interestingly, the low-temperature oxidation is later interrupted by multiple flashes of hot flame with a temperature below $T_m = 1800$ K. In the course of such a process, the droplet diameter $D$ continuously decreases (see Figure 10), whereas the $D^2/D_0^2$ ratio decreases almost linearly, as could be expected for the burning droplet. Accordingly, the rate constant of the low-temperature oxidation of the droplet increases with time and reaches its maximum value $k \approx 0.5$ mm$^2$/s by nearly the end of the process.

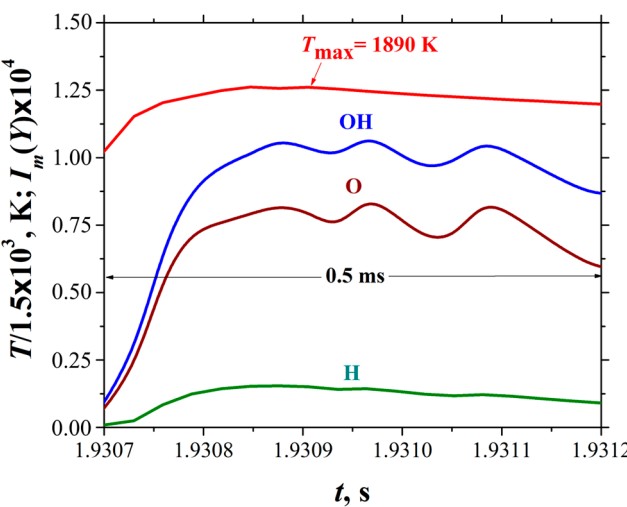

**Figure 11.** A fragment of calculated time histories of the maximum gas temperature, $T_m$, and the integral mass fractions of active species OH, O, and H at the forced ignition and combustion of the n-dodecane droplet with $D_0 = 2.8$ mm and $L = 0.55$ cm in air at NPT conditions.

As mentioned earlier in this paper, the thickness of the igniting layer significantly affects the evolution of the droplet. Thus, at $L = 0.54$ cm, the predicted number of flashes of hot flame decreases compared to the case of $L = 0.55$ cm, as does the average temperature between flashes. Accordingly, the resulting droplet burning rate constant becomes somewhat smaller. Figure 12 shows the case with $L = 0.53$ cm. After the extinction of the hot flame, the low-temperature reaction mode first appears at $T_m \approx 770$ K with a single stepwise increase in temperature up to $T_m \approx 950$ K. Thereafter, the maximum gas temperature exhibits several high-temperature flashes up to $T_m \approx 1750$–1500 K. Then, due to the accumulation of soot accompanied by radiation losses, the high-temperature hot flame is extinguished, and the maximum gas temperature drops to $T_m \approx 1100$–1150 K. The temperature range $T_m = T_l \approx 770$–1150 K is shown in Figure 12 as a blue band enveloping low-temperature flashes of cool and blue flame. In these conditions, the maximum rate constant of such a low-temperature oxidation process for the n-dodecane droplet is $k \approx 0.491$ mm$^2$/s.

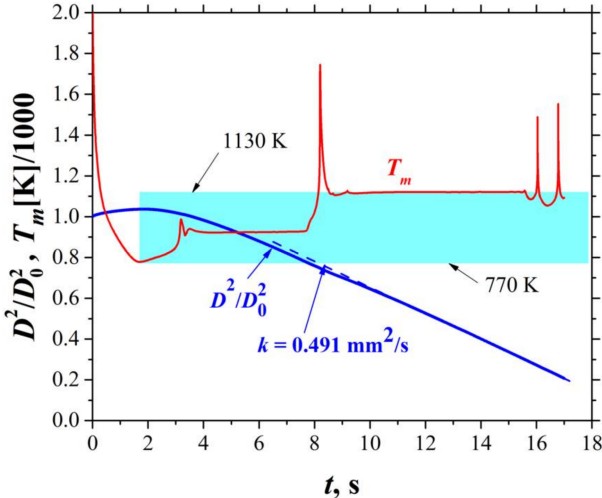

**Figure 12.** Calculated time histories of the maximum gas temperature, $T_m$, and the normalized squared droplet diameter, $D^2/D_0^2$, at the forced ignition and combustion of the n-dodecane droplet with $D_0 = 2.8$ mm and $L = 0.53$ cm in air at NPT conditions. The blue band envelopes the low-temperature flashes of cool and blue flames.

At $L = 0.52$ cm, the predicted number of flashes of hot flame decreases to a single flash at $T_m = T_l \approx 950$ K, whereas the rate constant of such a low-temperature oxidation of the droplet decreases to $k \approx 0.422$ mm$^2$/s. Finally, at $L = 0.51$, there is no ignition of the droplet. In this case, the maximum gas temperature $T_m$ drops to approximately 400 K, which persists for 30 s (the calculation time), while the droplet continues evaporating with a rate constant of $k \approx 0.00923$ mm$^2$/s. The constancy of $T_m$ with continued evaporation indicates a slow chemical process. The effect of $L$ on the burning/evaporation rate constant of the n-dodecane droplet is presented in Table 2. As seen, the calculated value of the burning rate constant of the n-dodecane droplet at NPT conditions depends on the ignition settings and ranges from $k = 0.422$ to $k = 0.500$ mm$^2$/s at hot flame ignition and attains a value of $k = 0.00923$ mm$^2$/s at a failure of hot flame ignition. The measured value of $k$ ranges from 0.378 to 0.398 mm$^2$/s [5,6]. The discrepancy between the calculated and measured values of $k$ can be attributed to the influence of the procedure of droplet ignition on the dynamics of further droplet combustion. The long-term effect of the ignition procedure on the evolution of gaseous spherical diffusion flames in terms of the soot yield was recently discussed in [35] with respect to the Flame Design—Adamant SE implemented recently at the ISS within the international NASA–Roskosmos collaboration. It was found that in the microgravity conditions, the soot yield in the gaseous ethylene diffusion flame is sensitive to the ignition up to a time of about 10 s.

**Table 2.** The effect of *L* on the burning/evaporation rate constant of the n-dodecane droplet at NPT conditions.

| *L*, cm | $T_l$, K | Number of Flashes | $k$, mm$^2$/s |
|---------|----------|-------------------|---------------|
| 0.55 | 900–1150 | Many | 0.500 |
| 0.53 | 770–1150 | 3 | 0.491 |
| 0.52 | 950 | 1 | 0.422 |
| 0.51 | - | None | 0.00923 |

It should be noted that the decrease in the droplet diameter during low-temperature oxidation occurs continuously in time rather than only during those short intervals when a cool flame appears as a result of the decomposition of alkyl hydroperoxide and/or a blue flame appears as a result of the decomposition of hydrogen peroxide. This is clearly seen in Figures 13 and 14 showing the evolution of integral mass fractions of alkyl hydroperoxide and hydrogen peroxide during the whole droplet lifetime at $L = 0.55$ (Figure 13) and 0.53 (Figure 14). The instants of $C_{12}H_{25}O_2H$ decomposition are marked by arrows. Each decomposition period lasts for a limited time less than 1 s. Note that, contrary to the curves $Y_i(t)$ in Figure 5 representing the evolution of $C_{12}H_{25}O_2H$ and $H_2O_2$ mass fractions everywhere in the homogeneous gas phase, the curves $I_m(t)$ in Figures 13 and 14 correspond to the instantaneous amounts of $C_{12}H_{25}O_2H$ and $H_2O_2$ accumulated in the entire space around the droplet, whereas, the cool, blue, and hot flames appear locally. Nevertheless, there is a good correlation between the evolution of the maximum temperature and peaks of the $I_m(t)$ curves. As seen from Figure 13, there are only three flashes of cool flame at the beginning of the droplet oxidation process, which correspond to the peaks of the $I_m(C_{12}H_{25}O_2H)$ curve. After each flash, the mass fraction of $C_{12}H_{25}O_2H$ drops to zero. As for the flashes of blue flame, their number is considerably larger, and they completely determine the later stage of the droplet oxidation process. Each flash of blue flame is accompanied by the abrupt decrease in the amount of $H_2O_2$. Contrary to $C_{12}H_{25}O_2H$, $H_2O_2$ is gradually accumulated in the space around the droplet up to the end of droplet lifetime.

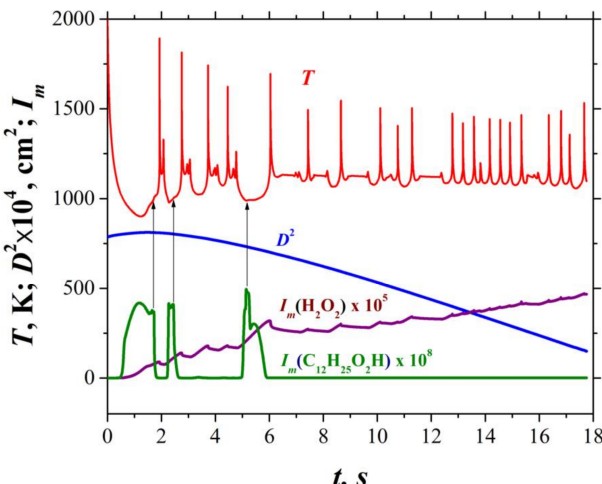

**Figure 13.** Calculated time histories of the integral mass fractions $I_m(C_{12}H_{25}O_2H)$ and $I_m(H_2O_2)$ as well as the maximum gas temperature, $T_m$, and the squared droplet diameter, $D^2$, at the forced ignition and combustion of the n-dodecane droplet with $D_0 = 2.8$ mm and $L = 0.55$ cm in air at NPT conditions. Arrows mark the instants of $C_{12}H_{25}O_2H$ decomposition.

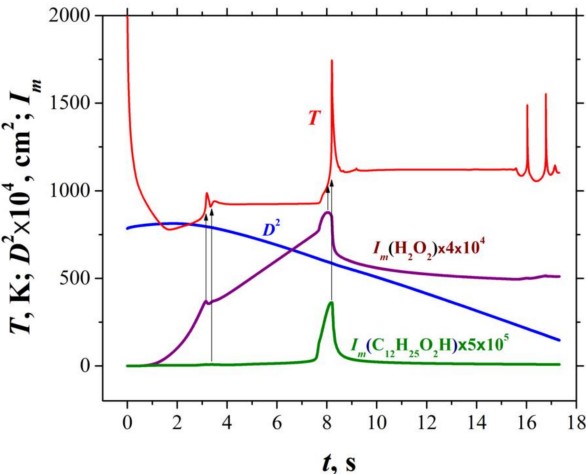

**Figure 14.** Calculated time histories of the integral mass fractions $I_m(C_{12}H_{25}O_2H)$ and $I_m(H_2O_2)$ as well as the maximum gas temperature, $T_m$, and the squared droplet diameter, $D^2$, at the forced ignition and combustion of the n-dodecane droplet with $D_0 = 2.8$ mm and $L = 0.53$ cm in air at NPT conditions. Arrows mark the instants of $C_{12}H_{25}O_2H$ and $H_2O_2$ decomposition.

### 3.4. Self-Ignition of the Droplet

The low-temperature oxidation of a hydrocarbon droplet can be also obtained at droplet self-ignition [29]. Let us consider self-ignition and low-temperature oxidation of an n-dodecane droplet with a small initial diameter $D_0 = 0.7$ mm in air at $T_{g0} = 643, 700, 750$, and 800 K and $P = 1$ bar. Figure 15 shows the calculated time histories of the maximum gas temperature, $T_m$, and the normalized squared droplet diameter, $D^2/D_0^2$, calculated for these conditions. The evolution of the maximum gas temperature corresponds to the multistage self-ignition of the droplet. First, a cool flame appears with a temperature of $T \approx 840$ K. Thereafter, a wave of the blue flame appears with a temperature of $T = 1000–1100$ K. Finally, a flash of hot flame occurs with a temperature jump up to $T \approx 1970$ K. The burning rate constant of the droplet in such a low-temperature oxidation process can be determined from the slope of $D^2/D_0^2(t)$ curve. Its value appears to be very close to the values obtained for forced ignition. At $T_{g0} = 643$ K (curve 1 in Figure 15), an interesting mode of droplet self-ignition is observed in the calculations. During the droplet lifetime, there is only cool and blue flame oxidation of fuel, i.e., there is no hot flame.

In this mode, the reaction zone is located very close to the droplet surface (at about one droplet radius), and fuel vapor reacts incompletely: it is gradually accumulated in the gas phase around the droplet. This effect was earlier reported in [29] for the low-temperature oxidation of n-heptane droplets.

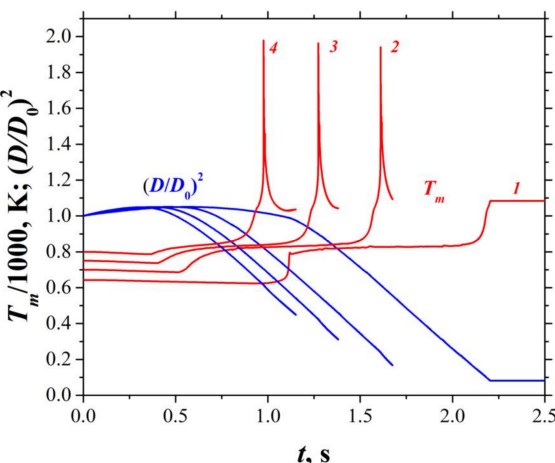

**Figure 15.** Calculated time histories of the maximum gas temperature, $T_m$, and the normalized squared droplet diameter, $D^2/D_0^2$, at self-ignition of the n-dodecane droplet with $D_0 = 0.7$ mm in air at $T_{g0} = 643$ (curve 1), 700 (2), 750 (3), and 800 K (4), and $P = 1$ bar.

Figure 16 shows the calculated time history of the maximum gas temperature, $T_m$, for the self-ignition of a large n-dodecane droplet with an initial diameter $D_0 = 4.0$ mm in air at $T_{g0} = 700$ K and $P = 1$ bar. It is seen that after the extinction of the first hot flame with $T_m \approx 1870$ K, the oxidation process with multiple temperature flashes from 1100 to 1200–1500 K is established. A detailed analysis of the calculation results shows that the flashes arise due to the decomposition of hydrogen peroxide with the release of hydroxyl, i.e., due to multiple blue flames, as was the case with the forced ignition of n-dodecane droplets (see Figures 13 and 14). This was supported by special mathematical experiments. Thus, on the one hand, if at some point in time, one takes the rate constant of hydrogen peroxide decomposition to zero, then flashes stop (see the blue curve designated as $A_{H_2O_2+M\rightarrow OH+OH} = 0$ in Figure 16). On the other hand, if at some point in time one removes the heat loss due to soot radiation in the model (see the blue curve designated as $S_{rad} = 0$), then blue flame flashes turn into a developed hot flame.

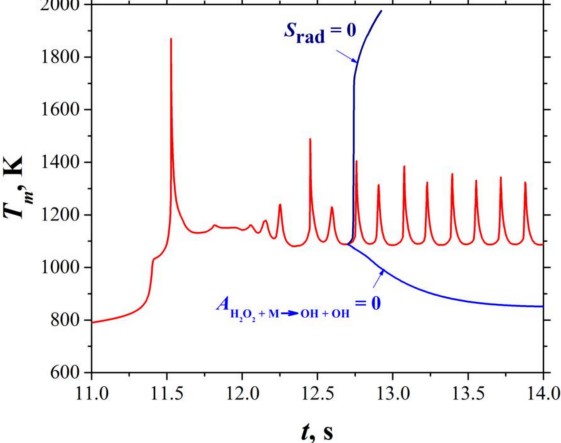

**Figure 16.** Calculated time histories of the maximum gas temperature, $T_m$, at self-ignition of the n-dodecane droplet with $D_0 = 4.0$ mm in air at $T_{g0} = 700$ K and $P = 1$ bar. Curves designated as $A_{H_2O_2+M\rightarrow OH+OH} = 0$ and $S_{rad} = 0$ show the flame temperature evolution in two mathematical experiments.

## 4. Conclusions

The mathematical model for simulating hydrocarbon droplet ignition, combustion, radiative extinction, and subsequent low-temperature oxidation with multiple flashes of cool, blue, and hot flames under microgravity conditions was applied to a single n-dodecane droplet. The model is based on the non-stationary differential equations of the conservation of mass and energy in liquid and gas phases with variable thermophysical properties. In the formulation of the problem, the property of spherical symmetry of all accompanying physical and chemical processes under microgravity conditions, as well as the concept of multicomponent diffusion in the gas phase, is used. The model is built for constant pressure in the gas–droplet system. An important advantage of the model is the use of a detailed kinetic mechanism for the oxidation and combustion of heavy normal alkane hydrocarbons, which, in particular, describes both multistage low-temperature oxidation with cool and blue flames and high-temperature combustion of n-dodecane. This mechanism is supplemented by a macrokinetic mechanism of soot formation, in which soot is represented by an equivalent gas with a molecular weight of carbon. Hot-wire ignition used in the space experiment is simulated in the calculation by placing a thin spherical layer with a high temperature around the droplet, leading to the appearance of a hot flame. Thus, the model takes into account multiple elementary chemical reactions in the vicinity of a droplet in combination with heat and mass transfer in liquid and gas, convection, heat release, soot formation, and heat removal by radiation.

The model allows reproducing the main features of the phenomena observed in space experiments. The main accomplishments are listed below.

(1)　Calculations confirm the important role of the soot shell and low-temperature reactions in the phenomenon of droplet radiative extinction with multiple flame flashes in the space experiment. In the vicinity of the droplet, even after the primary flame is extinguished, low-temperature chemical reactions, including chain exothermic reactions, can occur. Over time, active intermediate products are produced in low-temperature reactions, the main of which are alkyl and hydrogen peroxides. Upon reaching a certain critical concentration, the peroxides thermally decompose with the formation of hydroxyl radicals quickly reacting with other intermediates, causing a secondary flash of flame around the droplet. The secondary flash is followed by the formation of a secondary soot shell, which subsequently leads to the extinction of the flame and so on until the complete disappearance of the droplet. Under certain conditions, the intensity of the secondary flashes is very low that outwardly, the combustion of the droplet proceeds as flameless. Since the temperature of the gases in the vicinity of the droplet is higher than room temperature, the droplet evaporates much faster than in a room-temperature gas.

(2)　Calculations reveal the decisive role of the blue flame, arising due to the decomposition of hydrogen peroxide, in multiple flame flashes after the radiative extinction of droplet combustion.

(3)　Calculations with forced ignition of n-dodecane droplets reveal the effect of the ignition procedure on droplet evolution. It is shown that variation in ignition parameters changes the timing and the number of flashes of cool, blue, and hot flame.

(4)　The combustion rate constant of the droplet is also dependent on the forced ignition procedure, which correlates with recent findings in [35] on the long-term (about 10 s) effect of ignition on the soot yield in the gaseous spherical diffusion flame in microgravity conditions.

(5)　Calculations with droplet self-ignition reveal the possible existence of new modes of low-temperature oxidation of droplets without appearance of hot flame. In this case, the main reaction zone is located very close to the droplet surface and fuel vapor is oxidized only partly in it.

In general, space experiments CFI—Zarevo and Flame Design—Adamant provided new and valuable scientific information on the differences in the diffusion combustion of

droplets and gases on the ground and in microgravity. The fact that computer simulation reproduces the main features of the phenomena observed in the space experiments means that the physical and mathematical model possesses a predictive power and can be used to analyze various scenarios for the development of emergency situations with fires on manned spacecraft. More broadly, this model can be used to solve many other applied problems, such as choosing the operation conditions of a liquid rocket engine, jet engine, diesel engine, or a conventional steam boiler burning fuel oil to achieve a minimum soot yield, as well as choosing additives for fuel, providing the minimum soot yield during combustion.

**Author Contributions:** Conceptualization, S.M.F.; methodology, S.M.F. and V.Y.B.; formal analysis, S.M.F. and V.Y.B.; investigation, S.M.F. and V.Y.B.; writing—original draft preparation, S.M.F. and V.Y.B.; writing—review and editing, S.M.F.; supervision, S.M.F.; project administration, S.M.F.; funding acquisition, S.M.F. All authors have read and agreed to the published version of the manuscript.

**Funding:** This work was implemented within the framework of the Program of Fundamental Scientific Research of the Russian Federation "Processes of Combustion and Explosion", reg. No. 122040500073-4, and was funded by the state.

**Institutional Review Board Statement:** Not applicable.

**Informed Consent Statement:** Not applicable.

**Data Availability Statement:** The data will be available on request.

**Conflicts of Interest:** The authors declare no conflict of interest.

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
