# Peer review of "Simulation of Low-Temperature Oxidation and Combustion of N-Dodecane Droplets under Microgravity Conditions"

_fire, doi:10.3390/fire6020070_

Round 1
Reviewer 1 Report
Based on a typical unsteady droplet evaporation and combustion model, the self-ignition characteristics of droplets with different sizes under microgravity were studied. The results confirm that the important role of the soot shell formed around the droplet and low-temperature reactions in the radiation extinction phenomenon of the droplet under microgravity. The results of the droplet self-ignition reveal the possibility of a new mode of low-temperature oxidation of the droplet with the main reaction zone located very close to the surface of the droplet and with only partial conversion of fuel vapor. My comments are as follows:
1. Model validation. The model should be validated by experimental data.
2. Comparison between calculation and experimental results. This paper is to verify the experimental results conducted in ISS. There should be a comparison analysis between theoretical calculation and experimental results.
Author Response
We are grateful to the reviewer for valuable comments. We have made our best to follow all the comments. All changes in the revised manuscript are marked in yellow.
Based on a typical unsteady droplet evaporation and combustion model, the self-ignition characteristics of droplets with different sizes under microgravity were studied. The results confirm that the important role of the soot shell formed around the droplet and low-temperature reactions in the radiation extinction phenomenon of the droplet under microgravity. The results of the droplet self-ignition reveal the possibility of a new mode of low-temperature oxidation of the droplet with the main reaction zone located very close to the surface of the droplet and with only partial conversion of fuel vapor. My comments are as follows:
- Model validation. The model should be validated by experimental data.
- Comparison between calculation and experimental results. This paper is to verify the experimental results conducted in ISS. There should be a comparison analysis between theoretical calculation and experimental results.
To address these comments, we have added 3 new Figures (Figures 4 to 6) with the corresponding text to the manuscript. Figures 4 and 5 are related to the validation of the reaction mechanism, whereas Figure 6 is related to the validation of the droplet combustion model against ISS experimental data. In view of it, we renumbered all other figures.

Reviewer 2 Report
N-dodecane droplet ignition, combustion, radiative extinction, and subsequent low-temperature oxidation with multiple flashes of cool, blue, and hot flames under microgravity conditions is studied computationally in this article. The mathematical model takes into account multiple elementary chemical reactions in the vicinity of a droplet in combination with heat and mass transfer in liquid and gas, heat release, convection, soot for mation, and heat removal by radiation. Calculations confirm the important role of the soot shell formed around the droplet and low-temperature reactions in the phenomenon of droplet radiative extinction with multiple flame flashes in the space experiment at the ISS. This study has practical engineering significance and is recommended to be published.
Author Response
We are grateful to the reviewer for the kind review and valuable comments. Thank you.